# Enhancing Mobile "How-to" Queries with Automated Search Results Verification and Reranking

### Lei Ding
lding25@ucsc.edu
University of California, Santa Cruz
USA

### Jeshwanth Bheemanpally
jbheeman@ucsc.edu
University of California, Santa Cruz
USA

### Yi Zhang
yiz@ucsc.edu
University of California, Santa Cruz
USA

## ABSTRACT

Many people use search engines to find online guidance to solve computer or mobile device problems. Users frequently encounter challenges in identifying effective solutions from search results, often wasting time trying ineffective solutions that seem relevant yet fail to solve real problems. This paper introduces a novel approach to improving the accuracy and relevance of online technical support search results through automated search results verification and reranking. Taking "How-to" queries specific to on-device execution as a starting point, we developed the first solution that allows an AI agent to interpret and execute step-by-step instructions in the search results in a controlled Android environment. We further integrated the agent's findings into a reranking mechanism that orders search results based on the success indicators of the tested solutions.

The paper details the architecture of our solution and a comprehensive evaluation of the system through a series of tests across various application domains. The results demonstrate a significant improvement in the quality and reliability of the top-ranked results. Our findings suggest a paradigm shift in how search engine ranking for online technical support help can be optimized, offering a scalable and automated solution to the pervasive challenge of finding effective and reliable online help.

## 1 INTRODUCTION

When lacking the knowledge to complete a task, people usually rely on a search engine to retrieve potential instructions by asking "How-to" or similar style queries. These requests are "procedural queries" where the user's intent is to obtain step-by-step instructions for performing a specific task or operation. However, instructions extracted from retrieved results may not be executable, as the search engine primarily ranks pages based on the similiarty between the search query and candidate web pages [43]. Furthermore, although user engagement information, such as search logs and page quality indicators like PageRank, are utilized for ranking results in modern search engines, the optimal solution often varies on a user-specific basis. The use of different operating systems, screen types, and app versions motivates a personalized reranker. In addition, search

engines integrated with large language models (LLMs) often unavoidably involve the risk of generating inaccurate or fabricated information, commonly known as hallucinations [26].

How do users solve the problem? Typically, a user could visit multiple retrieved pages from a search engine triggered by a "How-to" query, which briefly describes their ultimate goal. If a page that looks relevant contains some step-by-step instructions, the user might try to follow the instructions to see whether it solves the problem. However, the potential issue is that they often experience frustrations and waste considerable time attempting various instructions across different pages. Can a search engine further assist the user more efficiently? Inspired by such time-consuming manual procedure, we endeavor to substitute it with an automatic instruction verification process as an additional component of the search process. Our hypothesis is that search results verification for technical "How-to" queries can achieve decent accuracy given the recent research progress on multimodal LLMs, especially with the help of state-of-the-art GPT.

In particular, we introduce a three-stage solution to verify the instruction set and rerank the relevant pages accordingly, as illustrated in Figure 1: The first stage involves an Instruction Extraction Model to obtain step-by-step instructions from each retrieved page for a "How-to" search query. The second stage verifies their quality by automatically simulating the instruction execution on devices. The third stage reranks retrieved results based on the execution information.

As an initiative, we have developed a comprehensive end-to-end solution to rerank "How-to" search query results on Android systems, selecting mobile applications from different domains. This can be enhanced as a platform-agnostic approach with slight adaptions to support other similar environments, such as web, iOS and desktop. Our solution enhances search engine results by reranking pages based on simulated verification on Android mobile devices, and our preliminary experimental outcomes have demonstrated the effectiveness of our methods. Our key contributions are outlined as follows:

(1) We propose adding search result verification and reranking into the search process, relieving users from tedious manual verification.
(2) For technical "How-to" queries, we proposed a three-stage solution. In the first stage, we propose an information extraction solution to extract step-by-step instructions from web pages using generative AI with grounding techniques. In the second stage, we propose and build out a generic action agent to take instructions in natural language format and

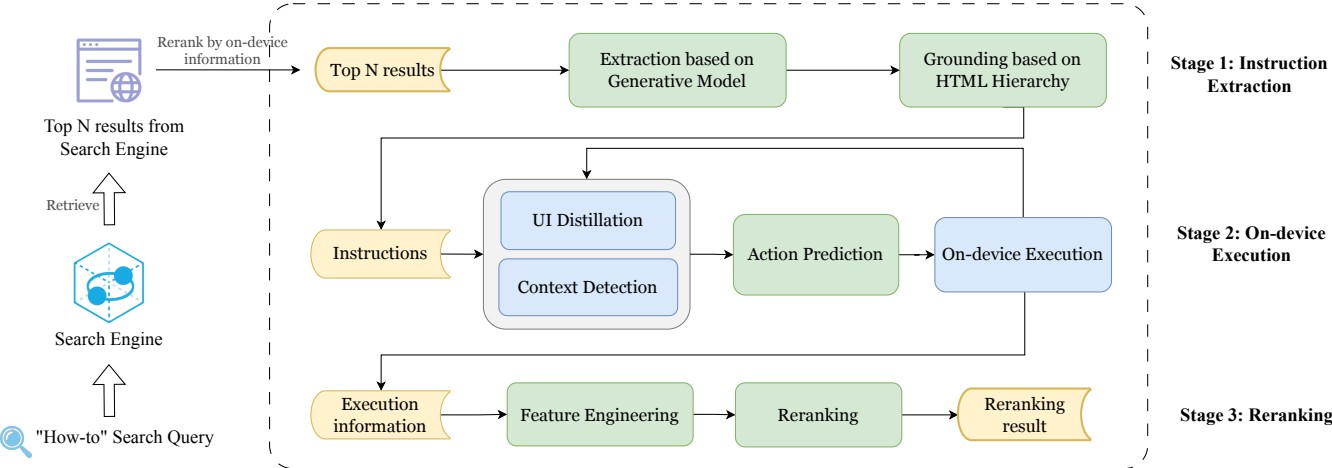

**Figure 1: Overview of our three-stage system to rerank web pages based on on-device execution verification**

execute actions on a client device. In the third stage, we introduce a range of features based on the execution outcomes, and further utilize them to rerank the retrieved results.

(3) To support future research in this direction, we developed a new research platform MagicWand, which facilitates the verification and reranking process for "How-to" queries of Android mobile applications. On top of it, we collected a new "How-to" WeWeb dataset with human annotators.

(4) We evaluated the proposed idea and the results show it significantly enhances the performance of a leading baseline search engine (i.e. Google).

## 2 RELATED WORK

As far as we know, there is no prior work on search results verification for "How-to" queries using simulated execution. Furthermore, the techniques we choose to use when developing different components of the proposed solution are motivated by existing works from multiple domains. Therefore, we briefly review three important domains that influenced the technical choices we made in our research.

### 2.1 Information Extraction

Information extraction from web pages [2], as a broader concept of our extraction task, was studied extensively back in the 1990s [10]. There are two methodologies: heuristics based approaches and machine learning based approaches. Heuristics approaches apply either a single metric or a set of rules along a hierarchy HTML tree to pinpoint the main content blocks. For instance, [52] considered "text density" and the DOM tree structure, while [42] segmented the HTML tree and then used heuristics for classification. In comparison, machine learning approaches in early studies involve sequence labeling methods and deep neural networks: [56] employed a hidden Markov model and a convolutional neural network to classify web regions; [25] used browser-rendered visual features to enhance model performance.

Recently, researchers also leveraged LLMs to facilitate relevant downstream tasks. For example, [18] enabled T5-based models for

semantic classification of HTML elements, description generation for HTML input, and autonomous web navigation. DOM-LM [8], MarkupLM [28] used transformer-based architecture to represent the DOM tree and further improve classification accuracy. A new dataset PLAtE of list-like websites, such as online stores, was developed by Amazon research team to analyze how LLMs work in structure extraction [49]. Researchers also endeavored to better utilize pre-trained LLMs to extract target information to avoid training LLMs. The instruction extraction approach used in our research is inspired by the prior work that used zero-shot GPT3 to extract plans from text [19].

### 2.2 Acting Agent

Enabling intelligent agents to behave according to instructions involves two main tasks: the first is empowering agents with sufficient knowledge to make decisions and act skillfully. The second is to execute actions, collect feedback, and recognize environmental changes caused by these actions. To achieve the first, researchers often leverage and enhance LLMs' reasoning and tooling capabilities [36]. For the second, agents should be enabled to interact with the host system to understand feasible actions and their results. Recent studies include using pre-trained GPT2 [45] in household simulations [30], RT1 [3] and RT2 [66] in robot control. This section will focus on how to adapt these tasks to the Android platform on which we will carry out our experiments.

***Acting Agent on Android:*** An early work on Android was [32] using LSTM to predict fixed action and parameter pairs on dataset RICO [7]. Later, Seq2Act [31] utilized Transformers for more flexible action execution given instructions. This approach set a strong baseline on a new dataset UGIF [55], further enhanced by META-GUI [53] with a proposed multimodal Action Prediction Model for conversational scenarios. Along this work, Auto-UI [62] built an "Action Chain" based model using UI representation extracted from BLIP-2 [27] as visual input, which outperformed two other competitive approaches (i.e. Behavior Cloning and Simple action Chain of Thoughts [58]) on the AWIT dataset [46]. However, they

overlooked the Android UI Control hierarchy, and transformers trained without such knowledge cannot accurately predict actions given ambiguous or sparse instructions. [21] demonstrated that LLMs equipped with domain knowledge could break down high-level tasks into mid-level plans. Researchers were also starting to explore multimodal LLM agents in Android automation, as evidenced by [59, 61]. Our Action Prediction Model aligns with this line of research, however, we've refined them in a more fine-grained manner by distilling the UI Control list from runtime metadata and prompting GPT4-V dynamically based on the context.

*Action Execution:* Usually, researchers resorted to two technologies to detect UI Hierarchy and take actions on Android devices: One is Android Debug Bridge (ADB) [15], such as [54, 61], the other is Accessibility Service API [16], like [47, 48]. ADB-based solutions, like AndroidEnv [54], adbutils [1], need additional connection settings for Android devices and the paired computer, which is unrealistic for end users during daily usage. In comparison, Accessibility Service-based solutions can be delivered as standalone mobile applications due to the general availability in most systems. Considering this benefit, we decided to utilize the Accessibility Service API to develop our execution proxy, which laid down a solid foundation for action execution.

## 2.3 Reranking

In information retrieval (IR), reranking is the adjustment of document rankings based on query-document relevance after the initial retrieval. An important evolution of reranking began with RankNet [4] in 2005, applying gradient descent in the process of learning a ranking function. LambdaRank [5] later refined this approach by optimizing the gradients of the model scores and addressing issues caused by discontinuous gradients. And LambdaMART, combining MART [11] with LambdaRank, achieved victory in the 2010 Yahoo! Learning-to-Rank Challenge. Subsequently, similar neural ranking models like DRMM [17] and Duet [37] emerged. Recently, [40] introduced a context-aware neural network for item relevance calculation, and [44] proposed DASALC, a neural LTR model yielding performance comparable to LambdaMART. This progress inspires our adoption of NeuralNDCG [41], which utilizes NeuralSort [14] for end-to-end, gradient-based stochastic optimization using a NDCG-based sorting operator.

## 3 METHODS

The flow chart in Figure 1 outlines the overall process of our proposal primarily by extracting instructions from Web pages and validating them through the execution on the device. The validation outcomes are then used to refine the search result rankings. The system can overcome some limitations of traditional ranking algorithms by incorporating user-specific information (device, OS version, application version, etc.) and collecting client-side verification results.

This is a three-stage process. Stage 1 is **Instruction Extraction**: we take a high-level goal for an Android application as a "How-to" search query to retrieve results from a search engine. Given the initial top N retrieved web pages, a generative AI model processes each document to generate a sequence of instructions. Then, a grounding module analyzes the structure of the HTML web pages

and further improves the quality of the extracted instructions. Stage 2 is **On-device Execution**: the Action Prediction Model reads the instructions extracted from stage 1 and predicts actions to be taken on the device. Next, an Android agent takes those actions to validate whether the instructions work, collecting the execution information simultaneously. This is an iterative process until the agent either completes the task or can't proceed further. Stage 3 is **Reranking**: representation features of each <page, query> pair are extracted from execution information and used for reranking pages for each query. In this stage, search results that help yield better execution results are assigned with higher priority during the ranking.

For simplification, we have utilized pre-trained GPT models in our implementations, which enables us to rapidly set up a pipeline that covers all three necessary components. Now we delve into the technical details of each component: Open Domain Instruction Extraction, On-Device Execution, and Reranking.

## 3.1 Open Domain Instruction Extraction

This stage has been divided into the generation and grounding phases, as shown in stage 1 of figure 1. Our approach allows us to extract instructions from any web page without collecting any training data, i.e. zero-shot learning [1].

Following [42, 52] using LLM for information extraction, we first prompt a LLM model to extract relevant instructions from a given web page, otherwise generate "none". In the generation phase, for a given search query $S$ and each candidate website $W_i \in W$, we pass in the prompt of the generative model the search query $S$, candidate web page title, and a cleaned HTML document. The cleaned HTML is generated using a parsing algorithm that removes extraneous information, such as javascript and header information, from the web page but still preserves the hierarchical structure of the HTML snippets.

Since generative models tend to paraphrase instructions or even generate text content unbounded with the original HTML, we also introduce a grounding mechanism to guarantee the generated instructions match the content in the HTML to minimize hallucinations in the generated content. For grounding, we first extract all text snippets $C[i]$, $i \in [0, ...N]$ from the cleaned HTML with their corresponding XPaths. The first instruction generated from LLM is matched with the closest candidate snippet $C[r]$, using a combination of Faiss semantic [24] similarity and rouge [33] similarity. A typical matching criterion has been illustrated in section 4.3.1. For each subsequent instruction, only snippets having similar XPaths to the previous grounded steps are considered as matched step candidates. This approach maintains the sequential order of extracted instructions from the original web page. If generated instruction(s) can be grounded to HTML snippets from the original web page, the relevant instructions can be used for the next execution stage.

## 3.2 On-Device Execution

The on-device execution agent is designed to directly execute instructions on an end-user device. Simulating the steps on a device provides more accurate judgements on whether the instructions

---

[1]By realizing the non-trivial effects caused by the hallucination issue, we have further refined the aforementioned strategy, making the side effect to the minimal level. More details will be covered in the discussion section, as well as our future work

can be completed using a specific device. In our research, three components are developed to achieve this goal with the workflow depicted in figure 2:

- *Runtime UI Context Detection Module* delivers UI context information for the current screen state in a JSON format. It contains instructions, the UI control hierarchy, potential actions, and necessary runtime metadata. This information serves as input for the Action Prediction Model and is also used to extract reranking features [53].
- *Multimodal Action Prediction Model* predicts the action and its required parameters (e.g. UI control index for click action, direction (up, down) for swipe, etc.) to complete an instruction with a chain of actions [65].
- *Execution Proxy Module* takes the predicted action and executes it on the client device.

The execution loop continues until either all instructions have been finished, or the Action Prediction Model can't determine the next action, or it reaches the maximum number of steps[2].

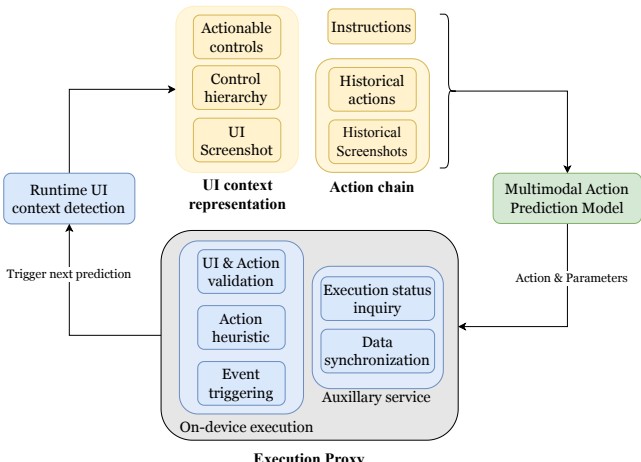

**Figure 2: Components and Workflow of On-device Instruction Execution Module**

*3.2.1 Runtime UI Context Detection.* This module collects data that characterizes the device's current User Interface (UI) context. The UI context consists of a UI screenshot and metadata of "actionable regions". An "actionable region" refers to a region on the screen where actions, such as clicking or scrolling, can be performed. These regions are hierarchically organized and can be associated with several control properties, like textual description, coordinate, scrollable, etc. For example, a region could be a navigation bar, with a back button and title as embedded child regions. To simplify, we currently focus on leveraging the textual description of the control directly associated with an "actionable region", like title, text description, etc.

Sometimes, a region may not directly contain any textual description, but its child regions often possess meaningful textural

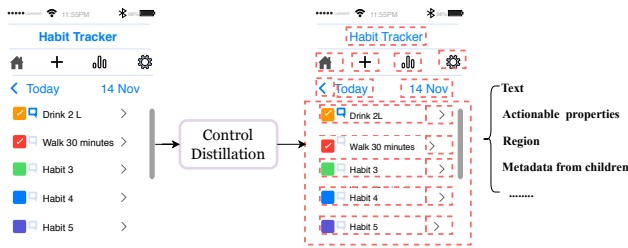

**Figure 3: Control distillation for On-Device Execution**

information, which is crucial for accurate action prediction but is unfortunately missing. We designate a control distillation algorithm to fill in this missing information, depicted in figure 3. First, invisible or non-actionable regions are filtered out. Then, descriptions from the nearest non-actionable child regions are consolidated hierarchically to represent the missing information.

*3.2.2 Multimodal Action Prediction Model.* Following [53, 61, 65], the Action Prediction Model prompts a multimodal LLM, GPT4-V [38, 60] in our current setting, where the current UI context and multimodal action chains [3] are passed as input to choose the target action from a set of *candidate actions* [4], such as click, swipe, input, back, etc.

For timestep $t \epsilon [1...T]$, GPT4-V will predict the next action given a prompt $P_t$ constructed using the following parameters:

- $I_t$: UI context for the current screen.
- $S_{1:t-1}$: List of screenshots of UI screens from previous actions.
- $A_{1:t-1}$: Action description for each previous action, including the action name and the chosen actionable region description.
- $C_t$: Set of possible actions based on available properties of distilled controls in $I_t$.
- $Context_t$: Information not covered by instructions but required for agent execution. For instance, a specific username and password are needed, when the instruction is "enter user name and password when open WeChat".

*3.2.3 Execution Proxy.*

*On-Device Execution Module:* It takes a predicted action and its parameters returned from the Action Prediction Model, locates the action to the corresponding region, and finally executes it. Considering a mobile application context can change dynamically, we introduce a heuristic to validate the predicted control associated with the target action and elegantly find a suitable alternative when mismatched. For flexibility, We also designate a fallback mechanism for candidate actions, so some of them can adjust slightly given different application settings. For instance, a click action can be tried with a similar gesture under instant Message application settings.

---

[2]In our experiment, the maximum number was set to 28 to avoid an endless execution loop.

[3]That is history of UI screenshots paired with each action taken

[4]Injecting the UI context into the prompt can help model better determine the next action based on our primary experiment. We expect more accurate action prediction results, if we can use this information appropriately as an input to train or finetune the model.

*Auxiliary Service Module:* It synchronizes the execution information of the mobile agents to external modules, helping to analyze and monitor the execution progress. In addition, this module also implements analysis plugins to integrate with GPT4V and GPT4 to summarize and evaluate the status of each execution trajectory.

## 3.3 Reranking

Execution information, both textual and visual information (UI screenshot and corresponding actionable regions), has been collected as listed in table 1 and converted into a reranking feature vector.

To avoid unnecessary complexity initially, we now use statistical measures of execution information, and leave it as future work to explore more advanced feature generation techniques, such as time series transformer or feature learning.

**Table 1: Collected information for execution**

|  | Description |
|---|---|
| Search query | Goal description on Android phone for Apps |
| Web page | Top 20 Pages retrieved from search engine |
| Instructions | Step-by-step instructions from web page |
| Mobile screens | UI screen saved per step |
| Distilled control list | Distilled control list on screen per step |
| Full control list | Full control list on screen per step |
| Action & Parameter | Predicted action and its parameter per step |
| Action attributing | Which instruction drives model to take action |
| Runtime information | More runtime information (issue, start time, etc.) |

*3.3.1 Feature Design.* All textual and visual features are normalized to a float value between 0 and 1. Below is a brief description of them.

**Table 2: Reranking features**

|  | Id |
|---|---|
| **Textual Features** | |
| Query term ratio in instructions | 1 |
| Relevance between page and search query | 2 |
| Keyword ratio | 3 |
| **Visual Features** | |
| Instructions completion degree | 4 |
| Action term ratio in instructions (avg, min, max, var) | 5-8 |
| UI term matching ratio in instructions (avg, min, max, var) | 9-12 |
| Matched UI term frequency on UI (avg, min, max, var) | 13-16 |
| Relative position of the last matched instruction term | 17 |
| Moving distancing of instruction terms | 18 |

*Textual Features.* $F_1$: **Query term frequency ratio in instructions** measures the relative frequency of "How-to" search query terms in the extracted instructions I, refer to Equation 1.

$$F_1 = \frac{\sum_{q_i \in Q} tf(q_i, I)}{length(I)} \qquad (1)$$

Q is the set of tokens from the search query without stopword, where $tf(q_i, I)$ is the term frequency of token $q_i$ in the extracted instructions I.

$F_2$: **Relevance between page and search query** measures the degree of relevance between a web page and the corresponding "How-to" search query. We prompt GPT to generate a relevance score ranging from [0, 1] using the search query, extracted instructions from the page, and the page title.

$F_3$: **Average term frequency of common keywords in instructions**: This feature measures the average frequency of common keywords found in the extracted instructions. We have used instructions from about 1000 web pages from Google Android Help Website and identified a set of $K$ most common keywords, excluding stopwords.

*Visual Features.* $F_4$: "**Instructions completion degree**" measures the degree to which the extracted instructions have been completed. GPT4-V is prompted to return a completion score of [0, 1], using the history of UI screenshots for each action taken during execution and extracted instructions as input.

$F_5, F_6, F_7, F_8$: average, min, max, variance of **Action term ratio in instructions** measures the average, minimal, maximum, standard deviation of the percentage of instruction's keyword that appears in each action description during the entire execution.

$F_9, F_{10}, F_{11}, F_{12}$: average, min, max, variance of **Visible UI Term Ratio in instructions** measures the average, minimal, maximum, standard deviation of the percentage of the current screen UI control's visible texts appeared in the instructions during the entire execution. This metric indicates **the alignment between the current UI screen and the instructions**. For example, for user registration, instructions often contain reference fields like username, password, and gender. A higher correspondence of these terms with UI elements suggests that the agent is more likely to move forward along the instructions.

$F_{13}, F_{14}, F_{15}, F_{16}$: average, min, max, variance of **Distilled UI ratio in instructions** describes the average, minimal, maximum, standard deviation of the occurrence of distilled UI control text in instructions during the entire execution. These metrics reflect the likelihood of our execution model accurately selecting the appropriate UI control from all distilled options.

$F_{17}$ to $F_{18}$: These capture how much of each extracted step is executed by comparing the list of actions taken for a given step[5] to its step text.

$F_{17}$: **Relative Position of Last Matched Instruction Term** identifies all matched positions of **Action Description** in instructions and uses the maximum value to quantify how far our execution can move from the starting position of extracted instructions.

$F_{18}$: **The moving distancing of instruction terms** measures the moving length from the first matched instruction term to the last matched one, indicating the action coverage in instructions.

Due to the page limit, we leave the exact equations for calculating the heuristic features $F_5$ to $F_{18}$ in our open-source website (TBA).

*3.3.2 Reranking Models.* In this section, we will illustrate which candidate pages have to be processed for reranking features and the details of reranking models.

*Verified Pages.* When the system fails to extract instructions or those pages do not contain any instruction, a candidate page does

---

[5]One step in a plan may require multiple actions to complete

not advance to the on-device execution stage. After that, our reranking modules prioritize all verified pages using a heuristic. In our experiments, a page is considered as **verified** only if it progresses to stage 2 and satisfies the following criteria: the execution proxy must successfully execute at least one step and the predicted instruction completion rate is non-zero. All other Pages are kept in their original relative order (i.e., not reranked) and put after the verified pages.

*Models.* Without losing the generality, three different learning-to-rank(LTR) models have been trained to rerank the verified pages using the aforementioned features.

The first model is a simple pairwise LTR logistic regression model, which optimizes the relative order of each pair of pages. Another two types of models are multilayer perception (MLP) transformer models, short for **TMLP**, using two different loss functions. One is LambdaLoss [57], a pairwise loss function. For each pair of items, the lambda value represents the change in the overall ranking metric, if the order of the two items were swapped. The lambda values are used to weight the errors in the pairwise comparisons, thus this loss function will penalize misorderings more if they have a higher lambda value. Another is NeuralNDCG loss [44], a method directly optimizing NDCG metric. This method introduces a NeuralSort operator based on a differentiable function (softmax) to approximate the non-differentiable ground-truth sort operator.

$$NeuralNDCG_k(\tau)(s, y) = N_k^{-1} \sum_{j=1}^{k} [scale(\widehat{P}) \cdot g(y)]_j \cdot d(j) \quad (2)$$

where $N_k^{-1}$ is the maxDCG at k-th rank, scale($\cdot$) is Sinkhorn scaling, g($\cdot$) and d($\cdot$) are their gain and discount functions.

## 4 EXPERIMENTS

Experiments have been designed to assess whether the proposed method enhances the retrieval performance of state-of-the-art search engines. We used Google as our original search engine throughout the whole process. Our experiments are hosted on MagicWand, a web platform that standardizes the training, execution, and evaluation pipeline for intelligent agents running on Android devices. It can spin up multiple Android emulators or connect with real devices on the server, which can be controlled by the user on a web interface or ADB-equivalent command interface.

### 4.1 Test Data: "How-to" WeWeb dataset

A new "How-to" WeWeb dataset has been collected using MagicWand, which covers instruction extraction, on-device execution, and data annotation for reranking. This dataset is used to measure the end-to-end reranking accuracy.

Its finalized version contains queries and instructions for 17 different apps spanning 9 categories of the Google Play Store and an additional "System" domain for the default system app [6]. We selected a single "How-to" search query in the System domain. For other domains, we recruit 6 students to propose about ten "How-to"

search queries per app. Using these queries, we retrieved valid web pages from the top 20 Google search results[7].

**Table 3: Statistics of "How-to" WeWeb dataset**

| Domain | Application | Queries | Pages | Instructions |
|---|---|---|---|---|
| System | Settings | 1 | 20 | 12 |
| Entertainment | Pluto TV | 11 | 202 | 9 |
| | YouTube | 12 | 240 | 26 |
| Education | Coursera | 10 | 200 | 2 |
| | Quizlet | 10 | 200 | 9 |
| Food & Drinks | DoorDash | 10 | 200 | 4 |
| | McDonalds | 10 | 200 | 2 |
| Communication | Google Chat | 10 | 200 | 9 |
| | Messenger | 12 | 240 | 27 |
| Shopping | Target | 10 | 200 | 1 |
| | eBay | 10 | 200 | 4 |
| News | FlipBoard | 10 | 199 | 11 |
| | BBC News | 10 | 200 | 1 |
| Maps&Navigation | Google Maps | 10 | 200 | 45 |
| | Here WeGo | 11 | 220 | 2 |
| Travel | Trip Advisor | 10 | 200 | 4 |
| | Expedia | 10 | 200 | 0 |
| | Total | 167 | 3321 | 168 |

We hired Amazon Mechanical Turk workers to extract relevant instructions from candidate web pages for "How-to" queries. Multiple workers analyzed each page to ensure data quality, requiring consensus on the extraction results to be labeled as positive. The resulting data, detailed in Table 3, includes statistics on the number of search queries, result pages, and pages with valid instructions for each domain.

*Human Verification Information:* With search queries and instructions extracted (firstly by humans and further verified by the extraction module) from pages, tasks were set up on MagicWand to record and access the execution of instructions on Android devices. 16 workers read each instruction, executed the instructions manually, and labeled whether the assigned "How-to" queries could be completed. Out of all pages evaluated, 168 pages are labeled as successful (i.e. positive) and 64 queries were associated with at least one positively labeled page. These human labels are used as the ground truth $y$: it is assigned with a value of 1 if a human annotator can follow the instructions on the page and complete the task, otherwise 0.

We captured the actions performed by the workers, along with their annotations, and saved them into documents with a pre-defined format. These documents included screenshots, JSON files that detailed the history of actions taken, and screen recordings capturing the entire interaction sequence. Additionally, a specialized plugin integrated with the platform was employed to extract the UI context.

---

[6]Based on Google result, the default system app category has higher proportion of web pages with instructions in top 20 result. In comparison, other categories vary for instruction proportion and therefore deserve more attention

[7]For the "Entertainment" domain, there are 18 search results that didn't have any valid web page; For the "News" category, 1 search result wasn't a valid web page.

## 4.2    Evaluation Metrics

Common reranking metrics are used: mean reciprocal rank (MRR) of the top one retrieved documents(MRR@1), precision of the top one and five retrieved documents (P@1 and P@5) and normalized discounted cumulative gain (NDCG) of the top five retrieved documents (NDCG@5).

## 4.3    System Configurations

### 4.3.1    Extraction.

*Generative model:* The state-of-the-art LLM, **GPT4** [38], is used to extract instructions from a candidate web page in our experiment, following [50, 51]. We set the temperature parameter as 0 to encourage direct extraction with minimizing generative content. We also design a prompt containing a list of valid and relevant instruction descriptions. For comparison, we have also tested cohere [9] using the same prompt, which could output accurate citations from the input web page, and found no noticeable performance difference compared with GPT4.

*Grounding:* A rouge score of above 0.7 or faiss similarity [24] score of under 0.25 is used to match a HTML text snippet of a webpage to a generated step.

### 4.3.2    On-Device Execution.

*Device setup.* On MagicWand, we have reserved an Android emulator to run tasks given instructions and collect execution statistics [8]. For efficiency, a batch-supported execution engine based on ad-butils [1] is also prepared to run multiple tasks on different devices simultaneously.

*Multimodal action prediction model.* To avoid invalid action parameters and keep exploring capacity, GPT4-V parameter "temperature" is set to 0.3 together with "max_tokens" of 300. We specify instructions, historical actions and UI screenshots in the prompt and adjust the prompt according to distilled UI controls.

*Execution proxy.* The UI context representation data with the execution status are saved into the central storage with a JSON index file describing the execution and associated resources. A GPT4V python plugin was developed to evaluate the execution information and generate the task completion rate for each execution.

### 4.3.3    Pre-training Reranking Models.

*Synthetic dataset for pre-training.* As the "How-to" WeWeb dataset is small, we decided to use it for testing; thus, the reranking is a zero-shot learning problem. We created another synthetic data to pre-train pairwise LTR models. The synthetic pre-training data - "How-to" META-GUI dataset was created based on META-GUI [53], a multimodal dataset for task-oriented conversational agents running on Android. The original data set includes 1125 conversations, 4684 dialogue turns, and 18337 Android execution trajectories. It covers six apps: weather, calendar, search, taxi, restaurant, and hotel mobile apps. This dataset has been used to generate a new "How-to" Meta-GUI dataset by the following process:

(1) Given 62 few-shot samples(the paired instruction, execution, search query), GPT3.5 is prompted to summarize an execution trajectory with its dialog turn to generate a list of step-by-step instructions and a relevant "How-to" search query. These are positive examples.
(2) Additional positive examples are created by swapping similar search queries in the same app domain from the generated samples of step one.
(3) Negative examples are created by identifying trajectories that fail to complete.
(4) We randomly modify or delete actions in a given trajectory from examples of step one to create further negative samples.

Table 4 summarizes the number of queries, executions, and instructions of our "How-to" META-GUI dataset.

**Table 4: Statistics of "How-to" META-GUI dataset used for LTR pre-training**

|                   | Query | Executions | Instructions |
|-------------------|-------|------------|--------------|
| Positive examples | 7884  | 13516      | 4684         |
| Negative examples | 944   | 25764      | 4684         |
| Total             | 8828  | 39280      | 4684         |

This synthetic dataset significantly differs from real data and the testing set. However, we expect possessing some form of pre-training material, even if not ideal, is preferable to having none at all, particularly in the zero-shot learning settings. We split *"How-to" META-GUI dataset* into training, validation, and test sets, then train the LTR models with the most suitable hyperparameters. Next, the trained models are applied to *"How-to" WeWeb dataset.*

*Reranking model setting.* Regarding the model details, **TMLP** is a fully connected layer of 96-dimensional input without normalization, followed by two transformer blocks with 384-dimensional features and a dropout rate of 0.1. A post-processing model takes the output of the transformer blocks and uses a linear layer to output the reranking score. For NeuralNDCG loss, its temperature is set as 1.0 and we use a "powered relevancies" gain function $2^x - 1$, where x is the relevance score. For LambdaRank loss, we set its parameters $\mu$ = 10, and $\sigma$ = 1.0. They are all optimized in the same training setting: an Adam optimizer with an initial learning rate of 0.001 is applied with a learning rate scheduler of size 50 and a decreasing $\Gamma$ rate = 0.1. Both training epochs and early stopping patience parameters are set to 20 with NDCG@5 as the validation metric.

## 4.4    Methods compared

To verify the model performance, we compared the following methods and calculated each query's MRR, P@1, P@5 and NDCG@5, using $\boldsymbol{y}$ as the relevance score.

- **Oracle**: This method consistently ranks the correct pages before the incorrect pages, which gives the theoretical optimal performance.
- **Baseline Google**: We order pages with/without extracted instructions by their Google rank and put the group with relevant instructions ahead of those without.

---

[8]Realistically, multiple emulators with various settings better simulate realistic cases. However, one device can help simplify our initial work and device maintenance.

- $F_4$-**based rank**: A single feature $F_4$ (i.e. instructions comple-
tion degree predicted by GPT4-V) is used to rerank pages
with extracted instructions.
- **LR**: This method uses a logistic regression-based LTR model,
pre-trained on the synthetic dataset.
- **NeuralNDCG + TMLP**: This method uses a transformer
with multilayer perception for LTR, pre-trained using Neu-
ralNDCG loss [44] on the synthetic dataset.
- **LambdaLoss + TMLP**: This method uses a transformer with
multilayer perception for LTR, pre-trained using Lambda
loss [57] on the synthetic dataset.

## 5 EXPERIMENTAL RESULTS

### 5.1 Comparison with Baselines

The performance metrics of different models are shown in Table 5.
The "Oracle" model has the highest performance across all metrics,
which is expected as the ideal benchmark. The baseline (i.e. Google)
has the lowest performance, showing that all other models have
improved the original Google ranking when taking execution status
into consideration. The improvements over the baseline are statis-
tically significant on LR, NeuralNDCG + TMLP, and LambdaLoss +
TMLP models. Despite their zero-shot learning nature, the results
demonstrate adding verification and reranking into the retrieval
process can significantly enhance the performance of a leading
baseline search engine.

In comparison, $F_4$-**based rank** performs worse than other rerank-
ing models. This shows reranking using **a strong yet single fea-
ture is feasible but not good enough**. Two neural ranking models
(TMLP) are better than LR probably because they can capture fea-
ture interrelationships. However, the difference is not very big,
possibly due to the low quality of the synthetic data used for pre-
training.

**Table 5: Performance on Test Data**

Results are performance on "How-to" WeWeb dataset using zero-shot
learning. *LR*, *NeuralNDCG + TMLP* and *LambdaLoss + TMLP* are
statistically significantly better than the baseline for *P@1*.

| Model | MRR | P@1 | P@5 | NDCG@5 |
|---|---|---|---|---|
| Oracle | 0.3353 | 0.3353 | 0.1365 | 0.3353 |
| Baseline: Google | 0.1782 | 0.1138 | 0.0850 | 0.1692 |
| $F_4$-based rank($F_4$) | 0.2107 | 0.1737 | 0.0946 | 0.1971 |
| LR | 0.2406 | 0.1976 | 0.1006 | 0.2227 |
| NeuralNDCG + TMLP | 0.2566 | 0.2275 | 0.1006 | 0.2275 |
| LambdaLoss + TMLP | 0.2594 | 0.2335 | 0.1030 | 0.2350 |

### 5.2 Further Analysis and Ablation Studies

Despite the promising performance on the zero-shot setting, we
want to further explore the correctness based on our data setting
and in which cases the system stumbled. In this section, we conduct
a further analysis to answer several critical questions.

*Q1. When do our proposed methods work and fail? Why?* To
answer it, success and failure cases have been analyzed to measure
the failure rate at each stage. After the instruction extraction stage,

**Table 6: Instruction completion degree analysis**

| Pages | Mean | Std | Min | 25% | 50% | 75% | Max |
|---|---|---|---|---|---|---|---|
| With instructions | 0.8647 | 0.2438 | 0 | 0.75 | 1 | 1 | 1 |
| No/Partial instructions | 0.3697 | 0.3218 | 0 | 0 | 0.5 | 0.5 | 1 |

398 pages (11.98%) have extracted instructions from the total 3321
pages, with 117 queries (about 70.05%) out of the total 167 queries.
Among the 398 pages, 75 pages (18.8%) with 39 queries(33.33%) con-
tain relevant instructions, while the remaining 323 pages with 108
queries are negative samples. At this stage, **LLMs' hallucination
issue** is a major factor: generative content unaligned with original
page content induces noises, which decreases the percentage of
true positive pages. We will further analyze how HTML grounding
has impacted extraction performance in Q2.

At the execution stage, the execution engine filters 119(29.90%)
pages from 68(58.12%) queries, and only 279 pages from 91 queries
are left. For the remaining pages, if the execution engine can com-
plete **more than one instruction** or the instruction completion
degree $F_4$ of the corresponding execution **is greater than 0**, the
extracted instructions are considered executable and are advanced
to the next stage. Among those dropped, 18(15.12%) pages have no
action taken by the execution proxy, and 101 (84.88%)pages have a
$F_4$ score of 0. Considering **the result of the former is promising**,
we focus on an analysis based on $F_4$: clearly shown in table 6, about
25% of pages with no or partial instructions are set with $F_4 = 0$ and
about 70% makes the execution engine struck at the first instruc-
tion (the engine can't move forward due to the missing necessary
information). Another good indicator is that about 75% pages with
instructions can support the execution engine to finish more than
75% instructions, which is aligned with human verification. How-
ever, top 25% pages with no or partial instructions are mistakenly
marked as $F_4 = 1$, introducing significant noise to reranking. This
highlights the need to incorporate more robust, execution-derived
signals in future work.

In the re-ranking stage, we analyze the relationship between
features and reranking scores for the best-performed models. Most
successful cases occur when models identify strong signals among
input features, such as $F_4$ and $F_2$, and accurately learn non-linear
weights to reflect on their interrelationship. However, current mod-
els tend to minimize negative signals even those with abnormal
values: for instance, a feature vector containing several very low
values, like 2.8e-05, will still receive a higher score just if it includes
multiple features with higher values. We attribute this to insuffi-
cient negative examples of training and validation sets or limited
features incorporated in reranking.

*Q2. What's the performance of the extraction module? Does
grounding matter for extraction? How instruction extract
works and how it impacts the final results (failed due to ex-
traction failure):* We use page URLs collected in the "How-to"
WeWeb dataset to fetch the corresponding web pages and apply
instruction extraction on individual pages to compare whether the
extracted content matches the ground-truth steps. Accuracy is used
to evaluate the instruction extraction performance, which measures
whether the model appropriately extracts steps or outputs "none"

**Table 7: Extraction Results Grounding vs None**

| Extracted Steps | Has relevant instruction | With grounding | Without grounding |
|---|---|---|---|
| Full Extraction | True | 78 | 100 |
| Partial Extraction | True | 18 | 42 |
| Full Extraction | False | 303 | 953 |
| No Extraction | True | 73 | 27 |
| No Extraction | False | 2849 | 2199 |
| Total Pages | | 3321 | 3321 |

under different cases. Extraction accuracy is listed with and without step grounding (Table 7). The results show that instruction extraction without grounding (full or partial) is $100 + 42 = 142$ out of 169 web pages with relevant instructions. 42 pages with instructions were partially extracted. However, it also falsely extracted instructions from 953 pages that don't contain relevant instructions. This happens because the model can hallucinate, extract the wrong instructions from a web page with relevant and irrelevant instructions, or significantly paraphrase the instructions.

With grounding, many partially extracted instructions are corrected by adding missing steps on the same XPaths on the HTML DOM tree, removing hallucinated instructions. As a result, it greatly reduced the number of false positives (i.e., extracted instructions from pages without relevant instructions) from 953 to 303, meanwhile reducing the number of true positives (i.e., extracted instructions from pages with relevant instructions) from $100 + 42$ to $78 + 18 = 96$.

***Q3. Does the zero-shot LTR for reranking help?*** We expect LTR will work better if we have training data with a similar distribution as the test data. However, considering the huge difference between the pre-training and testing data, is it still valid to have an LTR component? To clarify this question, we tried to rerank using a simple rule: promoting pages with $F_4 = 1$ (i.e. GPT4-V predicts the instructions are 100% completed) to the top without changing their related rankings. This rule-based reranking reaches $P@1 = 0.1317$, much worse than our proposed LTR approaches while not significantly better than the baseline. This suggests LTR models are more useful than a simple rule based on one strong feature.

***Q4. Which features contribute more to the overall performance?*** During the reranking performance analysis, we found that $F_4$ significantly influenced the final ranking: LTR models tend to assign a higher ranking score when $F_4$ values are higher. This tendency is more obvious when $F_4$ and $F_2$ are aligned. We attribute it to the limited diversity of our synthetic training data, leading to ranking decisions being dominated by strong features. Another interesting observation is that the remaining features, which focus on instruction, UI, and action alignments, fail to address some common but tricky circumstances in agent execution, resulting in incorrect ranking decisions. For instance, agents may get trapped in "execution cycles" where robots revert to a previous state due to ambiguous instructions. Despite this, the visual features and LTR models still tend to assign higher scores to these instructions. This underscores the necessity of incorporating more diverse training data and improving feature representation.

***Q5. How sensitive the model performance is for different LTR model settings?*** We tried Sigmoid, Tanh activation functions in the TMLP models to observe whether a non-linear function will impact the final result. Despite a little improvement, there is nothing statistically significant. We expect the small size of our test dataset and the big gap between training and testing data to make the results less sensitive to the structural perturbations on TMLP. The LTR models we trained are far from optimal, and better training data could improve the performance.

## 5.3 Discussion and Future Work

Our study represents a novel approach for search reranking. However, it is still preliminary and comes with several limitations. Therefore, it is necessary to examine those limitations and highlight our future directions.

***Sample size, scope and experimental constraints:*** One limitation is that it has been mainly covered on the Android platform, and other platforms, such as iOS, desktop and web [9], should be studied in the future, considering their potential difference with Android, especially Accessibility API. Although we test diverse mobile application domains, the coverage is not exhaustive. Our study also shows that the exact performance (P@1, etc.) is domain-specific. This may affect the generalization of our findings, and the reader should consider this when interpreting the results. When applying similar techniques to other platforms or domains, the exact number (P@1 etc.) would differ.

***Extension of instruction extraction:*** Although generative LLMs offer a quick start, the hallucination issue has heavily hindered the instruction extraction model, as noted by Ji et al. [23]. However, during further analysis, we have found that such issue heavily relies on how we represent HTML data to LLMs and how data verification has been organized. In our recent experiment, we have improved the algorithm for extracting cleaned HTML and including it in the prompt with a clear XML tag, making LLMs recognize it as a code snippet instead of unstructured text. Together with the long context support of gpt4-turbo [12], gpt4-o [13], a majority of instructions can be extracted successfully in the generative phase. In the grounding phase, we have reorganized the context in which the extracted instructions have been verified by the gpt4 models. We split the grounding phase into two subphases: the first phase focuses on the relevance between instructions and the given goal, and the next phase reexamines whether the visual representation of instructions in HTML page is aligned with the goal. Through these refinements, we achieved a better result on our current dataset, which can serve as a more promising baseline in our future work. Looking forward, we would like to replace those two-phase solutions with a single LLM-extraction model: to be specific, we want to integrate generative LMs with multimodal document models such as XDoc [6], MarkupLM [29], LayoutLMv3 [22], and LayoutLLM [35], aiming to enhance in-model extracting and grounding efficacy.

---

[9]We're now attempting to build a cross-platform execution proxy, following the design mentioned in the section 3.2.3. The key idea is to leverage platform-native API to capture accurate control hierarchies and expose UI-level interactions so that the technical concept in this paper can be naturally extended.

***Refinement of execution module:*** A more tightly coupled integration of the UI context and the Action Prediction Model can further enhance the execution performance. Moreover, our current prompt-based method, which used the actionable regions as input, does not effectively represent semantic groups on the UI layer. For example, an input box and its accompanying label, intended for entering a username, should be semantically grouped as a single region. Another follow-up is to enhance the previous work [20], [34] to improve the UI contextual information passed to the Action Prediction Model. In our recent work, by using Dino v1 [63] to understand UI segmentation and injecting into the UI contextual information, the execution module performs slightly better as knowing UI semantic groups. However, in the long term, a more promising direction is to follow the line of Ferret v2 [64] by embracing the fine-grained UI representation learnt by Dino v2 [39] into the vision part of the vision language model.

***Reranking model:*** Our current reranking model relies on simple feature engineering and does not use real-world training data. At the next step, we expect to acquire more real-world training data through strategic sampling. Rather than relying on manually engineered features, we could represent verification information, particularly the sequence of action screens and instructions, using more SOTA representation learning techniques. Using more realistic data and more advanced representation learning, we anticipate a substantial increase in the effectiveness of our proposed approach.

***Safety issues:*** Although verification agents help users avoid tedious manual verification, they might accidentally take harmful actions due to some misleading online content. In practice, we need to provide safeguards to prevent these risky actions from being taken by the execution agent. Ideally, a client environment that doesn't impact the user's real environment should be used. If not, we might allow each user to review and approve extracted instructions or at-risk actions before the automated execution. Another approach is to run server-side verification offline and display metadata about verification in the search results (e.g., "verified for app1 v3 on android v11"), which allows personalized search results without realtime verification on the client side.

## 6  CONCLUSION

We propose reranking top retrieved results for "How-to" search queries by promoting candidate web pages with verified executable instructions, which adds an additional layer of usefulness to a traditional search engine workflow. It ranks search results based on the actual completeness of candidate instructions, besides traditional metrics such as textual relevance and authority scores. The experimental results demonstrate our approach could further improve a very strong baseline search engine(i.e. Google).

This paper is a pioneer work in improving the reliability and usefulness of search results of online help resources for "How-to" queries about software tasks. Along this direction, several promising future research and applications are on the horizon, like adapting the proposed workflow in figure 1 to support verification of other platforms (web, macOS, Windows, iPhone, etc.), or collecting user feedback to improve the reranking models and also improving

the software agent's ability to interpret and execute a wider range of instructions.

## ACKNOWLEDGMENTS

For the "How-to" WeWeb dataset, we would like to appreciate our research collaborators: Sijia Zhong, Valentina Tang, Zoey Zhou, Jimmy Chen, Hanwen, Yang, and Chole Wong. Without their efforts, it would have been impossible to finish data collection on time. They also provided valuable suggestions and feedback to MagicWand. Additionally, we would like to thank Yaxuan Wang for her effort to refine the figures used to illustrate key ideas concisely and elegantly. Finally, we would like to thank our early contributors: Annabelle Miin, Anastasia Miin, Krish Pai, Rithvik Chavali, and Saket Pathak. They joined our research at the very beginning and worked tirelessly on data collection and verification. Their efforts encompassed sampling search queries, selecting relevant Android apps, and ensuring data accuracy through a rigorous two-stage verification process.

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
