# OpenReview forum: "Enhancing Mobile "How-to" Queries with Automated Search Results Verification and Reranking"
_ACM.org/SIGIR/2024/Workshop/Gen-IR — Gen-IR_SIGIR24_

### Official Review · Reviewer_BGbn · 2024-05-23
**This paper introduces an interesting approach to enhance mobile "how-to" queries with automatic search results verification and reranking,**

**Rating:** -2
**Confidence:** 3

**Review:**

This paper introduces an interesting approach to enhance mobile "how-to" queries with automatic search results verification and reranking, aiming to improve user experience. In particular, this work proposes a three-stage solution: extracting step-by-step instructions from web pages, verifying these instructions by simulating their execution on Android devices, and reranking the search results based on the success of these executions.

Strength:

1. the proposed approach is novel.
2. The 3-stage system design is well-designed and the authors detailed the steps very well.
3. The authors conducted a comprehensive evaluation of the proposed techniques across various domains.
4. The research questions proposed are well proposed.

However, while reading this paper, I also have the following doubts:
weakness：
1. Some of the claims and hypotheses in this paper are unsupported. e.g.
- The authors made the hypothesis in Introduction Section that: search results verification for technical "how to" queries can achieve decent accuracy given the recent research progress on multimodal LLMs.., what is the definition of "decent accuracy? In addition, the authors may pose the challenge faced before the multimodal LLMs for solving the introduced issue.
- Also, the authors claim : "GPT4 has been used to extract instructions from a candidate web page.", but didn't give any reference paper as evidence.


2. Among the various features proposed, I am more curious about which type of feature would contribute more to the overall performance?

3. the presentation of this paper has a large room to be improved. For example:
- in Section 4.3.1, state-of-art, should be "state-of-the-art".
- there should be a space before the citation bracket in many places.
- Please unify the usage of "multimodal" or "multi-modal"

---

### Official Review · Reviewer_vcee · 2024-05-27
**Enhancing Mobile "How-to" Queries with Automated Search Results Verification and Reranking**

**Rating:** 1
**Confidence:** 4

**Review:**

* Novel Approach: The paper introduces an innovative method to enhance the accuracy and relevance of online technical support search results through automated verification and reranking of "How-to" queries. It leverages AI to interpret and execute instructions in a controlled environment, which is then used to rerank search results based on the success of the solutions.
* Comprehensive Evaluation: A detailed assessment of the system is provided, demonstrating significant improvements in the quality and reliability of top-ranked result. The findings suggest a new direction for optimizing search engine ranking for technical support, offering a scalable and automated solution to a common challenge.
The paper's strength lies in its potential to transform the way users find effective and reliable online help, making it a significant contribution to the field of information retrieval.
* Weakness:
  * Hallucination Issue: The generative LLMs used for instruction extraction may produce hallucinated content not aligned with the original page, leading to noise and false positives.
  * Platform Limitation: The study primarily covers the Android platform, and the applicability to other platforms like iOS, desktop, and web needs further exploration.
  * Domain-Specific Performance: The performance metrics like P@1 are domain-specific, which may limit the generalization of the findings.
  * Execution Module Refinement: The integration between the UI context and the Action Prediction Model could be improved for better execution performance.

---

### Official Review · Reviewer_eUMS · 2024-05-27
**Search results verification and reranking on mobile device**

**Rating:** 2
**Confidence:** 3

**Review:**

This paper propose to utilize automated search results verification and reranking to improve the accuracy and relevance of online results. They proposed a three-stage solution for technical "How-to" queries and experiment results show its effectiveness.

---

### Decision · Program_Chairs · 2024-05-31

**Decision:**

Accept

**Comment:**

The paper proposes a three-stage solution to improve online search results through automated verification and reranking. The reviewers acknowledge the novelty of the proposed approach and the comprehensive evaluation results. For the camera-ready version, the paper could benefit from addressing some questions raised by the reviewers, such as improving presentation and citations, and providing more discussion about limitations in the existing framework, such as hallucination in instruction extraction and platform limitations.